# Diamine Groups on the Surface of Silica Particles as Complex-Forming Linkers for Metal Cations

**DOI:** 10.3390/molecules28010430

**Published:** 2023-01-03

**Authors:** Veronika Tomina, Nataliya Stolyarchuk, Olha Semeshko, Mariusz Barczak, Inna Melnyk

**Affiliations:** 1Department of Chemisorption and Hybrid Materials, Chuiko Institute of Surface Chemistry NASU, 17, Generala Naumova, 03164 Kyiv, Ukraine; 2Department of Physical and Physico-Chemical Methods of Mineral Processing, Institute of Geotechnics SAS, 45, Watsonova, 04001 Kosice, Slovakia; 3Research Sector, Kherson National Technical University, 24, Beryslavske Highway, 24, 73008 Kherson, Ukraine; 4Institute of Chemical Sciences, Faculty of Chemistry, Maria Curie-Sklodowska University, 20031 Lublin, Poland

**Keywords:** sol–gel method, ethylenediamine groups, polysiloxanes, polysilsesquioxanes, metal ion adsorption

## Abstract

Novel spherically shaped organosilica materials with (propyl)ethylenediamine groups were obtained via a modified one-pot Stöber co-condensation method. The porosity of these materials was tuned with the controlled addition of three silica monomers acting as structuring agents (tetraethoxysilane and bridged silanes with ethylene and phenylene bridges). The morphologies and structures of the synthesized materials were studied by SEM, DRIFT spectroscopy, CHNS elemental analysis, low-temperature nitrogen adsorption–desorption, and electrokinetic potential measurements. Their sizes were in the range of 50 to 100 nm, depending on the amount of structuring silane used in the reaction. The degree of the particles’ agglomeration determined the mesoporosity of the samples. The content of the (propyl)ethylenediamine groups was directly related with the amount of functional silane used in the reaction. The zeta potential measurements indicated the presence of silanol groups in bissilane-based samples, which added new active centers on the surface and reduced the activity of the amino groups. The static sorption capacities (SSCs) of the obtained samples towards Cu(II), Ni(II), and Eu(III) ions depended on the porosity of the samples and the spatial arrangement of the ethylenediamine groups; therefore, the SSC values were not always higher for the samples with the largest number of groups. The highest SSC values achieved were 1.8 mmol_Cu(II)_/g (for ethylene-bridged samples), 0.83 mmol_Ni(II)_/g (for phenylene-bridged samples), and 0.55 mmol_Eu(III)_/g (for tetraethoxysilane-based samples).

## 1. Introduction

Heavy metals are well-known environmental pollutants due to their toxicity and ability to bioaccumulate in the bodies of living organisms at various stages of the food chain [1]. The negative changes in plants [2], fish [3], mammals [4], and humans [5] have been widely reported. In order to prevent large concentrations of metals from entering water bodies, it is necessary to organize the proper disposal of metal-containing wastes and implement efficient water treatment facilities at factories and plants. Along with ecological aspects (e.g., water purification), the economic aspects (e.g., recovering valuable metals for their reuse) play an important role in the design of suitable solutions for the environmental management of heavy metals’ presence in the biosphere.

Limited mining possibilities and the increasing costs for various metals, such as copper, nickel, and europium [6,7,8], mean that the prospect of their recovery from industrial waters and wastewater as well as deposited wastes seems to be a promising strategy to meet the growing demand for these heavy metals. Sorption is considered to be one of the most desired approaches for recovering various metals from industrial and natural waters because of its effectiveness, ease of use, low cost, and environmental sustainability [9]. The development and application of adsorption processes is always considered along with the development of new adsorbents with well-developed porous structures, large surface areas, and high contents of functional groups [10].

Amino-containing organosilicas are known to be very good sorbents for the removal of metal ions from waters and wastewaters due to the complex-forming properties of amino groups [11,12,13,14,15,16] and the adjustable characteristics of the porous structure. In addition to sorption, amine-containing silicas are used as bearers in nanomedicine [17,18], and a one-step method for their preparation, taking into account the change in porosity and the nature of the surface, is an immediate challenge. Moreover, the sorption capacity of such materials can be improved by increasing the number of active centers. However, as our previous studies have shown, increasing the number of amino groups by increasing the proportion of functionalizing silane (3-aminopropyltriethoxysilane, APTES) in the reaction mixture leads to the formation of non-porous samples and the inaccessibility of the amino groups to interactions [19]. However, there are several other possible approaches: using a functional agent with a larger number of nitrogen donor centers that are connected to the surface by a hydrolytically stable ≡Si-C-bond or replacing the structuring agent. Taking into account previous research [20] and aiming to increase the number of surface adsorption centers, a silane with ethylenediamine functional groups, N˗[3˗(trimethoxysilyl)propyl]ethylenediamine, was applied as a functional agent [21]. However, there are several nuances in such syntheses. For example, Yokoi et al. [22] demonstrated that the lengthening amino-organic fragment improved the sorption properties of MCM-41 when prepared via direct co-condensation but did not when prepared using surface grafting. Hydrophilic amino-organic fragments were shown to interact with the hydrophobic chains of surfactants, influencing the formation of porosity and ensuring the uniform distribution of surface groups. However, when the co-condensation occurs in a strongly acidic medium, the protonated amines are incorporated inside the silica walls of SBA-15, and despite high nitrogen loading, do not have sites available for heavy metal adsorption [21]. Meanwhile, surface grafting was also shown to increase the sorption capacity of the materials with amino-organic motifs but only with the optimal loading of samples with amino-containing groups [23]. In addition, mesoporous materials can be obtained using other approaches and in the course of single-pot synthesis using bridging silanes [19]. Therefore, we tested three different silanes (tetraethoxysilane, bissilane with an ethylene bridge, and bissilane with a phenylene bridge) as structuring agents at various ratios. Thus, the aim of our study was to synthesize mesoporous spherical silica particles with high contents of ethylenediamine groups using a one-pot Stöber technique as well as to determine the parameters that affect the adsorption of copper, nickel, and europium ions and compare the obtained results with other similar amine-functionalized materials [14,24].

## 2. Results and Discussion

### 2.1. Features of the Synthesis of Spherical Silica Materials with Diamine Functional Groups

The (propyl)ethylenediamine-bearing samples were synthesized by applying the one-pot modified Stöber method in an ammonia medium using three different structuring agents, as schematically represented in Figure 1.

Polysiloxane samples **TNN2**, **TNN3**, and **TNN6** were obtained on the basis of TEOS. Polysilsesquioxane materials were synthesized with the following structuring silanes: samples **ENN1**, **ENN2**, and **ENN4** were made using ethylene-bridged BTESE, while samples **BNN1**, **BNN2**, and **BNN4** were made using phenylene-bridged BTESB. In all cases, TMPEDA was the source of (propyl)ethylenediamine functional groups onto the surface of the organosilicas. Among the goals of applying bissilanes as structuring agents was also creating mesoporosity in the emerging particles without the involvement of the template synthesis that is generally used to make mesoporous samples. It also should be mentioned that our synthesis method differed from the template method in the type of catalyst that was used, which was alkaline in our method. At the beginning of the process, the reaction was autocatalyzed by the amino groups of TMPEDA, while later the ammonium hydroxide was added. During syntheses with bridged silanes, fluoride ions were applied to accelerate the hydrolysis of their ethoxy groups. We chose an alkaline catalyst instead of an acidic catalyst for several reasons: (1) the acid as a catalyst could protonate amino groups, making them unavailable for interactions, and (2) acid promotes gelation and slows down hydrolysis. Thus, we sought to obtain spherical particles, preferably mesoporous, that would have a large amount of nitrogen-containing complex-forming functional groups on the surface, which would make it possible to use them in sorption processes. During the synthesis, the molar ratio of the structuring and functionalizing silanes was varied in order to adjust the porosity of the final materials, and the numbers after the letters in the names of the samples indicate this ratio. Previous experience with the aminosilicas [20] has shown that varying the proportion of TEOS or bissilanes in the reaction medium allows for the regulation of the porosity of the final materials.

### 2.2. Morphologies, Structures, and Structural-Adsorption Properties of (Propyl)ethylenediamine-Bearing Silica Materials

Generally, Stöber synthesis, which we applied, results in the formation of spherical silica particles, the size of which can be controlled, but the patterns are individual for each component of the reaction mixture. Thus, we analyzed the morphologies of the obtained samples using SEM, and the images are presented in Figure 2.

Comparing the microphotographs, it can be concluded that the synthesized materials were predominantly formed by spherical particles of various sizes in the range of 50–100 nm and that the morphologies of the samples varied, depending on the nature and the content of the structuring component. Polysiloxane samples **TNN3** and **TNN6** were formed by loosely packed spherical particles of ~100 nm in size. Meanwhile, polysiloxane sample **TNN2**, with a lower share of structuring silane TEOS, was a monolithic material formed by spherical particles less than 50 nm in size. The presence of an alkyl bridge (-CH_2_-CH_2_-) in the siloxane network (polysilsesquioxane samples **ENN1** and **ENN2**) contributed to the formation of particles with sizes of 50–80 nm and a relatively loose packing of spheres; meanwhile, an increase in the content of alkyl bridges in the network led to particle growth up to 100 nm (sample **ENN4**). The polysilsesquioxane with phenylene bridges that was synthesized at the equimolar ratio of structuring to functional silanes (sample **BNN1**) was characterized by a particle size of 50 nm. Meanwhile, a growing content of phenylene bridges in the network increased the polydispersity and heterogeneity of the materials. Consequently, polysilsesquioxane samples **BNN2** and **BNN4** did not consist of individual particles, and their sizes were difficult to determine; the particles formed smaller and larger agglomerates from 300 to 700 nm and even larger.

In order to characterize the porous structures of the materials that were produced and to consider the impact of the structuring silane components on the porosities of the samples, we measured and analyzed the isotherms of the low-temperature adsorption–desorption of nitrogen. They were used to calculate the values of S*_BET_*, V*_p_*, and PSD and to estimate the contributions of different types of pores to the total pore volume (Figure 3 and Table 1). It is known that the porosity of samples depends on the ratio of the reaction components as well as the size of the functional group. In these systems, the influences of both the above-mentioned factors were observed, leading to the formation of non-porous spherical silica particles at the lowest structuring to functional silane ratio for samples with TEOS and BTESE (small and flexible bridges). Samples synthesized with larger proportions of structuring silanes TEOS and BTESE were characterized by high specific surface area values. Comparing the data for three series of samples, it is obvious that an increase in the proportion of the structuring agent in the reaction medium contributed to the formation of a more developed porous structure.

It is interesting that TEOS-based polysiloxanes had high S*_BET_* values (Table 1). Such values can be due to the small sizes of the particles forming agglomerates with different tightness values. It is obvious that for the sample **TNN2** there were no gaps between the particles, for the sample **TNN3** they were at the level of mesoporosity, and for the sample **TNN6** there were larger gaps and slits between the particles. Indeed, according to the calculations (Figure 3a and Table 1), the porosity of the TEOS-based polysiloxane samples was due to the predominance of slit-like pores, which could be considered mesopores due to their sizes. The low-temperature nitrogen adsorption isotherms for sample **TNN3** (Figure 3a) belonged to type IV, in accordance with the IUPAC classification [25]. It should be noted that there were hysteresis loops in these isotherms, which were caused by capillary condensation in the mesopores. The isotherm of sample **TNN6** could be attributed to type I, as it did not have a hysteresis loop.

Regarding the ethylene-bridged samples, an increase in the contribution of the bridging silane improved the S*_BET_* and V*_p_*. This may have resulted from the formation of particles of different sizes and their heterogeneous packaging. Thus, sample **ENN2** had predominantly slit-like mesopores, similar to the TEOS-based samples, whereas sample **ENN4** already had gaps and a larger number of nanopores. This was reflected in the shape of the adsorption isotherms. The isotherm of the **ENN4** sample could be attributed to the Langmuir type.

In contrast, the phenylene-bridged samples were usually porous, regardless of the ratios of the reaction components, but sample **BNN2** was non-porous under these conditions. The isotherm of the **BNN1** sample belonged to type IV, with the characteristic hysteresis loop of mesoporous materials. This sample contained a majority of cylindrical mesopores (Table 1), which are usually formed between homogeneously packed uniform particles. The nitrogen adsorption–desorption isotherm of the **BNN4** sample could be attributed to the Langmuir type. The value of the specific surface area increased due to the presence of slit-shaped and cylindrical nanopores and mesopores.

Therefore, silane with ethylenediamine groups (TMPEDA), during the polycondensation with TEOS, contributed to the formation of smaller particles and the production of materials with well-developed surface areas and porosities compared to the silane with aminopropyl groups (APTES) [26]. However, the impact of TMPEDA on the porous characteristics of the polysilsesquioxane samples was not so evident [14].

### 2.3. Properties and Behavior of Surface Groups of (Propyl)ethylenediamine-Bearing Organosilicas

The introduction of functional groups is a means of giving the materials specific sorption properties, for example, towards metal ions. Therefore, it is important to conduct a qualitative and quantitative analysis of the contents of functional groups. Ethylenediamine functional groups can be identified in the DRIFT spectra (Figure 4) by the presence of two absorption bands of low intensity in the region of 3298 and 3378 cm^−1^, characteristic of the ν_s,as_(NH) stretches of amino groups, while the band at 1459–1458 cm^−1^ refers to δ(NH_2_). An intense absorption band at 1018–1157 cm^−1^, present in the spectra of all samples, refers to the asymmetric stretches of the ν_as_(Si-O-Si) network [27]. Bridging silanes are represented in the silica network by ethylene and phenylene structuring elements. Thus, the presence of -CH_2_- units from the ethylene bridge (as well as from the alkyl fragments of the functional groups) is manifested in the IR spectra of xerogels as a group of absorption bands in the region of 2812–2935 cm^−1^, corresponding to the symmetric and asymmetric stretches of the C−H bonds as well as bands of low intensity at ~1265 and 1416 cm^−1^ that can be attributed to their bending vibrations. The presence of –C_6_H_4_– bridges can be identified by the groups of absorption bands around 3008–3055 and 1500–1932 cm^−1^ in the DRIFT spectra. There are also absorption bands at 3726 and 3644 cm^−1^ in the spectra of polysilsesquioxane materials (especially phenylene-bridged materials), referring to the OH vibrations of silanol groups. In summary, characteristic absorption bands in the DRIFT spectra indicated the presence of a siloxane network, organic bridges in the composition of the bridged samples, some silanol groups, and functional amino-containing groups that were introduced during the synthesis.

The analysis of the CHNS elements indicated that the content of nitrogen (and hence amino groups) in the samples of adsorbents naturally increases with increasing portions of the functional agent TMPEDA (and decreasing portions of structuring agents) in the reaction mixtures. Specifically, the number of functional groups for polysiloxane samples was 1.95–3.40 mmol/g, the number for BTESE-based polysilsesquioxanes was 1.50–3.40 mmol/g, and the number for BTESB-based polysilsesquioxanes was 1.14–2.91 mmol/g. 

The electrokinetic potential arising at the phase boundary between the solid and liquid under the action of a direct electric field is widely used to characterize the electrical properties of a surface. The changes in this value depending on the pH indicate the nature of the groups on the surface. The results of studying the electrophoretic mobility of dilute suspensions of adsorbents in aqueous solutions in a wide pH range were recalculated into zeta potentials (Figure 5). The zeta potential values of the samples were within −20 mV < ζ < +20 mV in the pH range under study, which indicated the instability of such suspensions and the tendency of particles to agglomerate. In addition, all groups of samples had a general tendency to shift the isoelectric point to a more acidic environment with a decreasing number of nitrogen-containing groups (Table 2 and Figure 5). Such a shift may also be connected with an increasing number of silanol groups on the surface coming from bissilane polycondensation [28]. Consequently, due to the partial protonation of ethylenediamine groups and the practical absence of silanol groups (according to the DRIFT spectra), polysiloxane TNN samples were characterized by positive zeta potential values in the range of pH values from 2 to 9, and their isoelectric points were alkaline (pH = 9.32–10.77). Samples of ethylene-bridged polysilsesquioxanes had acidic isoelectric points (pH = 3.03–4.59). The isoelectric points of phenylene-bridged polysilsesquioxanes shifted from alkaline (pH = 9.19) to acidic (pH = 4.11) with an increasing number of structuring agent moieties (and, indirectly, silanol groups) and a decreasing (propyl)ethylenediamine group content.

### 2.4. Study of Sorption Properties of (Propyl)ethylenediamine Spherical Silica Materials with Respect to Metal Ions of Cu(II), Ni(II), and Eu(III)

The ethylenediamine functional groups of the synthesized organosilicas were expected to interact with heavy metal ions, in particular, the Cu(II), Ni(II), and Eu(III) ions attested in the current research. Their adsorption isotherms and linearized graphs are plotted in Figure 6, while the parameters are given in Table 3.

According to the SSC values (Table 3), the ethylenediamine-bearing samples with the highest affinity to Cu(II) ions can be ranked as follows: **ENN1**—1.8 mmol/g (C_NN_ = 3.4 mmol/g) > **TNN6**—1.79 mmol/g (C_NN_ = 1.95 mmol/g) > **ENN2**—1.62 mmol/g (C_NN_ = 2.36 mmol/g) > **BNN2**—1.38 mmol/g (C_NN_ = 1.88 mmol/g). 

The samples, with the highest affinity to Ni(II) ions can be ranked as: **BNN1**—0.83 mmol/g (C_NN_ = 2.91 mmol/g) > **TNN2**—0.72 mmol/g (C_NN_ = 3.4 mmol/g) > **TNN3**—0.69 mmol/g (C_NN_ = 2.91 mmol/g) > **ENN1**—0.46 mmol/g (C_NN_ = 3.4 mmol/g). 

The samples, with the highest affinity to Eu(III) ions can be ranked as: **TNN3**—0.55 mmol/g (C_NN_ = 2.91 mmol/g) > **BNN1**—0.52 mmol/g (C_NN_ = 2.91 mmol/g) > **ENN1**—0.40 mmol/g (C_NN_ = 3.4 mmol/g) > **ENN2**—0.37 mmol/g (C_NN_ = 2.36 mmol/g).

Thus, there was no direct dependence between the number of functional groups and the SSC values of the samples. Therefore, the particularities of cation adsorption should be analyzed in terms of the features of the structure and group availability for each separate type of organosilica. Taking into account the sizes of the ionic radii, namely Cu^2+^ = 73 pm, Ni^2+^ = 69 pm, and Eu^3+^ = 106.6 pm [29], we compared their adsorption on different types of samples within groups that were prepared from the same synthesis components but at different structuring to functional silane ratios. 

#### 2.4.1. Sorption Peculiarities of TNN Samples

As described earlier, there was a direct dependency between the amino group content and the quantity of TMPEDA used in the synthesis of the TNN samples. Specifically, the content of the (propyl)ethylenediamine groups in TEOS-based polysiloxanes decreased with a reduced proportion of TMPEDA silane in the reaction mixture from 3.4 to 2.91 and 1.95 mmol/g. However, the SCC towards Cu(II) ions was inversely dependent on the C_NN_ values and increased from 1.04 to 1.06 and 1.79 mmol/g with a reduced proportion of TMPEDA silane in the reaction mixture. A similar pattern was observed earlier for polysiloxanes with 3-aminopropyl groups (from APTES) on the surface [26]. This behavior can be explained by the limited availability of functional groups due to back-bonding, preventing their interactions with target ions, especially if the groups are not discretely located on the surface [30]. Moreover, the availability of groups can be affected by the porosity of the samples: porous polysiloxane silicas **TNN6** and **TNN3** showed the highest values of SCC to Cu^2+^ with their lowest contents of functional groups (Table 3). Comparing the correlation coefficients of Cu(II) adsorption on polysiloxane TNN samples with the Langmuir and Freundlich isotherm models, it can be concluded that adsorption occurs on a heterogeneous surface. At the same time, both the Langmuir (R^2^ = 0.972) and Freundlich (R^2^ = 0.976) isotherms were almost equally suitable for **TNN3**. The formation of a monolayer was evident here due to the adsorption of copper(II) ions on different types of adsorption centers. The fact is that in this case, during the formation of a six-coordinated Cu^2+^ complex, the composition of the complex can include from 1 to 4 N atoms [31], depending on the arrangement of the groups. However, when the groups are located at a distance, the composition of the complex is simplified and the molar ratio of the number of groups to the amount of metal becomes 1 to 1, as for sample **TNN6**, and in general, the sorption capacity is higher. Other places in the complex are occupied by water molecules.

Contrary to copper(II), the adsorption of nickel(II) ions, which have a smaller radius, directly depended on the C_NN_ functional group content. However, polysiloxanes **TNN2** and **TNN3** demonstrated similar SCC values towards Ni(II), regardless of the number of groups. That is, in the sample with the largest number of groups, some groups did not interact, and therefore the question of the feasibility of incorporating such a large number of groups arises.

Europium(III) ions, which are the largest in size and form eight-membered complexes, displayed the highest adsorption (SSC = 0.55 mmol/g) on **TNN3** polysiloxane. This material, **TNN3**, had a group content of 2.91 mmol/g, but it was also formed by loosely packed spherical particles and had the highest S*_BET_* of 427 m^2^/g and a V*_p_* of 0.51 cm^3^/g (93% consisting of mesoporosity). In addition, comparing the correlation coefficients, the adsorption data were better fit by the Freundlich isotherm model, meaning that the interaction occurred on a heterogeneous surface. Meanwhile, the lowest adsorption of Eu(III) ions among TNN polysiloxanes was shown by the **TNN2** sample with the highest content of functional groups (C_NN_ = 3.40 mmol/g). This sample, **TNN2**, despite the highest C_NN_ value, featured a monolithic material of densely packed ~50 nm particles with low SBET and poorly developed porosity. The Eu(III) ion adsorption pattern of sample **TNN6** (incorporating 43% and 33% fewer ethylenediamine groups than **TNN2** and **TNN3**, respectively, and with morphological and porous characteristics similar to **TNN3**) was similar to **TNN2**, both in the SSC value and the tendency to form a monolayer, following the Langmuir isotherm model. That is, in the adsorption of such large ions as Eu^3+^, the functional group content is important [32], but the location and the arrangement of these functional groups on the surface has a predominant role in the formation of the adsorption complex.

#### 2.4.2. Sorption Peculiarities of ENN Samples

For all the attested ions, Cu^2+^, Ni^2+^, and Eu^3+^, there was a direct dependence between the contents of functional (propyl)ethylenediamine groups and the amounts of metal cations taken up. That is, according to their SSC values, the samples could be ranked as **ENN1** > **ENN2** > **ENN4**. As shown by the experience with the APTES-functionalized samples [14,19,24], incorporating ethylene bridges in the structure of a polysiloxane network is a means of both introducing mesoporosity in the structure of the materials and improving the availability and reactivity of the functional groups. Supposedly, the same is true for the (propyl)ethylenediamine groups from TMPEDA. Ethylene bridges promote the isolated location of N-containing groups and, therefore, prevent them from having water-mediated hydrogen interactions with each other and so called ‘back-bonding’, making them available for interactions with target cations. All synthesized ENN samples adsorbed Cu^2+^ better, while the Ni^2+^ and Eu^3+^ SSCs were close in magnitude. According to the calculations (Table 3), the adsorption complex of Cu(II) usually contained from three to four atoms of nitrogen, i.e., one Cu^2+^ binds with two functional groups, forming a monolayer coating on the surface. The **ENN1** sample was an exception. It was characterized by the highest SSC towards Cu(II) ions, 1.8 mmol/g, and according to the correlation coefficients, the process of adsorption was better described by the Freundlich adsorption isotherm, indicating the heterogeneity of adsorption centers and the possible involvement of silanol groups. The same pattern was observed for Ni^2+^ uptake by **ENN1**: its isotherm was also better fit by the Freundlich adsorption model, confirming the surface heterogeneity, but the SCC value (0.46 mmol/g) was four times less than that of Cu^2+^, indicating a different structure for the adsorption complex. For the rest of the polysilsesquioxanes Ni(II) ions, uptake occurred at homogeneous adsorption sites with monolayer coverage of the surface.

The Eu^3+^ ion sorption by ENN samples was directly dependent on the functional group content. Apparently, the incorporation of ethylene bridges in the siloxane network promoted an arrangement of surface groups suitable for the emergence of optimal Eu^3+^/(propyl)ethylenediamine group complexes and the formation of a monolayer. Despite the difference in size compared to the Ni(II) ion, the adsorption complex of the Eu(III) ion on the surface of the sorbents included a similar number of nitrogen atoms and, accordingly, a similar metal/N atom ratio.

#### 2.4.3. Sorption Peculiarities of BNN Samples

It was expected that the incorporation of phenylene bridges into the siloxane network, similar to the ethylene bridges, would increase the reactivity of the (propyl)ethylenediamine functional groups by ensuring their proper arrangement on the surface and preventing ‘back-bonding’. However, the adsorption features of BNN polysilsesquioxanes were different from the ENN samples.

Concerning the adsorption of copper(II) ions, BNN samples do not have a large sorption capacity, but the formal ratio of one Cu^2+^ to two functional groups (3–5 nitrogen atoms) is generally preserved. Nickel(II) ions form aquacomplexes in solutions that stabilize them and slow down adsorption; therefore, the hydrophobicity of the phenylene bridges may affect the aquacomplexes of Ni(II) ions. In confirmation of this theory, the phenylene-bridged polysilsesquioxane sample **BNN1** had the highest Ni^2+^ SSC value of 0.83 mmol/g among all samples studied for the current article. 

Moreover, when comparing the dependence of the zeta potential on pH (Figure 5) and the adsorption data (Table 2), a general tendency (for polysiloxanes and polysilsesquioxanes) of decreasing Ni(II) ion adsorption with a shift in pI to lower pH values was observed. This shift in pI, which was especially noticeable for BNN samples, was connected with the increasing content of surface silanol groups, the presence of which may hinder the formation of Ni^2+^ adsorption complexes.

When analyzing the correlation coefficients, the sorption of Cu^2+^ and Ni^2+^ by **BNN1** and **BNN2** were better described by the Freundlich model, which indicates the heterogeneity of the surface. However, the magnitudes of the data correlation coefficients with the Langmuir model were quite close to the Friendlich model (except for Ni^2+^ and **BNN2**), so we can suppose the formation of a monolayer due to the interaction with amino groups simultaneously involving other surface groups (silanol or phenyl). Meanwhile, with an increasing proportion of phenylene-bridged silane in the structure (**BNN4**), the sorption centers were distributed more uniformly, ensuring a uniform distribution of ions that formed a monomolecular adsorption layer on the surface of the samples.

The research into the adsorption of Eu^3+^ ions by BNN samples once again confirmed the importance of both the functional group availability and arrangement on the surface on the emergence of the adsorption complex. The highest adsorption, SSC = 0.52 mmol/g, was observed for **BNN1** (C_NN_ = 2.91 mmol/g), a mesoporous sample formed by uniformly packed spherical particles 50 nm in size. Despite a quite high content of (propyl)ethylenediamine groups, C_NN_ = 1.88 mmol/g, **BNN2**, as a non-porous sample with a low specific surface area comprised of particle agglomerates, showed the lowest affinity towards Eu^3+^, 0.17 mmol/g. In all the cases, the correlation coefficients of the sorption data with the Langmuir adsorption isotherm model were higher than with the Freundlich model. This indicates the monomolecular adsorption of Eu(III) ions at uniform adsorption sites. However, phenylene bridges, contrary to ethylene bridges, appear to be too rigid and spacious to ensure the optimal arrangement of (propyl)ethylenediamine residues on the surface.

In general, an interesting relationship emerged: the sample with the highest sorption of copper(II) ions (**BNN2**) had the lowest sorption of Ni(II) and Eu(III) ions in the series, while the sample with the worst sorption of copper(II) (**BNN1**) demonstrated the highest uptake of Ni(II) and Eu(III) ions. This provides support for the role of silanol and phenylene residues in the interactions with the target cations.

In general, for all the obtained samples, both polysiloxanes and polysilsesquioxanes are best for taking up copper(II) ions from water. Meanwhile, the sorption of nickel(II) and europium(III) ions is worse. The growth of adsorption at the lengthening of the amino-organic fragment could only be observed in the case of copper sorption on the surface of TEOS-based polysiloxane particles (Figure 7), in accordance with other authors in [21]. The Cu^2+^ complexes with amine/polysiloxanes were shown to contain two amino groups [14]. Meanwhile, Cu complexes with ethylenediamine/polysiloxanes can also involve two nitrogen atoms in the coordination sphere, but they both may come from the same group. Therefore, such ethylenediamine/polysiloxane samples may demonstrate higher SSC values (Figure 7).

The Cu^2+^ complexes with amine/polysilsesquioxanes contain two amino groups [14], whereas Cu^2+^ complexes with ethylenediamine/polysilsesquioxanes may involve 2–4 nitrogen atoms in the coordination sphere. In addition, the particles’ aggregation can affect the SSC values of the samples, blocking some of the groups and preventing their interactions with target ions.

It also should be mentioned that the ratios calculated in Table 3 for 1Ni/xN and 1Eu/xN complexes are conditional because such numbers of nitrogen atoms in the coordination spheres are impossible. This means that a large number of groups did not participate in the interaction, which may result from being either too close or too distant relative to each other or the location at the entrance of the pore, which prevented the ions from passing inside the pore to bind with the groups [11]. This feature, namely the different sorption behaviors of the same sample in relation to different ions, is sometimes difficult to explain, but it can be used in the creation of selective materials for the removal and separation of certain cations in water.

## 3. Materials and Methods

### 3.1. Materials

The silanes used in the synthesis: N˗[3˗(trimethoxysilyl)propyl]ethylenediamine, (CH_3_O)_3_Si(CH_2_)_3_NH(CH_2_)_2_NH_2_ (TMPEDA, 97%, Merck), as a source of functional groups, and tetraethyl orthosilicate, Si(OC_2_H_5_)_4_ (TEOS, 98%, Aldrich); 1,2-bis(triethoxysilyl)ethane, (C_2_H_5_O)_3_Si-C_2_H_4_-Si(OC_2_H_5_)_3_ (BTESE 95%, J&K Scientific Ltd., Lommel, Belgium) or 1,4-bis(triethoxysilyl)benzene, (C_2_H_5_O)_3_Si-C_6_H_4_-Si(OC_2_H_5_)_3_ (BTESB, 96%, Aldrich), as structure-forming agents. An ammonium hydroxide solution (NH_4_OH, 25%, Lach-Ner) and an ammonium fluoride solution (1%, prepared from analytical-grade NH_4_F salt, Reachim) were also used as catalysts, while ethanol (C_2_H_5_OH, p.a., 96%, microCHEM) was used as a solvent.

The reactives for the adsorption studies: Cu(NO_3_)_2_·3H_2_O (99.5%, Merck); Ni(NO_3_)_2_·6H_2_O (chem.pure, Makrokhim, Ukraine); Eu(NO_3_)_3_·6H_2_O (99.9%, Alfa Aesar); NaNO_3_ (chem.pure, Makrokhim, Ukraine); NaNO_3_ (p.a., 99.5%, ITES s.r.o.); EDTA di-sodium versenate (0.1 mol/L, Constanal chelaton III, microCHEM); hexamethylenetetramine (p.a., 99.0%, microCHEM); xylene orange, and sodium salt (pure, Aeros organics). 

### 3.2. Synthesis

Polysiloxane samples: **TNN2** (TEOS/TMPEDA molar ratio = 0.020175 mol/0.010131 mol = 2/1), **TNN3** (TEOS/TMPEDA molar ratio = 0.0222416 mol/0.007472 mol = 3/1), and **TNN6** (TEOS/TMPEDA molar ratio = 0.0222416 mol/0.003736 mol = 6/1). Ethanol (100 mL), TEOS (4.5, 5, or 5 mL, respectively), and TMPEDA (2.2, 1.62, or 0.81 mL, respectively) were constantly stirred for 1.5 h. Then, 5 mL of a 25% NH_4_OH solution was added to the reaction mixture. The reaction was terminated after 20 h.

Ethylene-bridged polysilsesquioxane samples: **ENN1** (BTESE/TMPEDA molar ratio = 0.0025 mol/0.0025 mol = 1/1), **ENN2** (BTESE/TMPEDA molar ratio = 0.0025 mol/0.00125 mol = 2/1), and **ENN4** (BTESE/TMPEDA molar ratio = 0.0025 mol/0.000625 mol = 4/1). First, 25 mL of EtOH was mixed with 0.9 mL of BTESE and 0.2 mL of 1% NH_4_F, followed by the addition of TMPEDA (0.55, 0.28, or 0.14 mL, respectively) and 2 mL of 25% NH_4_OH. The reaction continued for 2–3 h.

Phenylene-bridged polysilsesquioxane samples: **BNN1** (BTESB/TMPEDA molar ratio = 0.0025 mol/0.0025 mol = 1/1), **BNN2** (BTESB/TMPEDA molar ratio = 0.0025 mol/0.00125 mol = 2/1), and **BNN4** (BTESB/TMPEDA molar ratio = 0.0025 mol/0.000625 mol = 4/1). First, 25 mL of EtOH was mixed with 0.9 mL of BTESB and 0.2 mL of 1% NH_4_F, followed by the addition of TMPEDA (0.55, 0.28, or 0.14 mL, respectively) and 2 mL of 25% NH_4_OH. The reaction continued for 1 h.

All samples were triple-washed with C_2_H_5_OH, centrifuged, and dried at 100 °C for 24 h. The samples were stored in a desiccator over anhydrous calcium chloride.

### 3.3. Methods

A JEOL JSM-6060 LA analytical scanning electron microscope was used to obtain scanning electron microscopy (SEM) images. A Costech Microanalytical Kelvin-1042 instrument was applied to record the low-temperature nitrogen adsorption–desorption isotherms of previously degassed samples (100 °C for 2 h in the helium flow) at −196 °C and to calculate the BET specific surface area (S*_BET_*) and sorption pore volume (V*_p_*). The SCV/SCR method, which is a self-consistent regularization (SCR) procedure applied to the integral adsorption equations based on a complex model with slit-shaped/cylindrical pores and voids between spherical particles packed in random aggregates (SCV model), was practiced to evaluate the pore-size distributions (PSD) in the materials, as described elsewhere [33]. A Thermo Nicolet Nexus FTIR spectrometer was operated to collect diffuse reflectance infrared Fourier transform (DRIFT) spectra of the samples ground with solid KBr in the 400–4000 cm^−1^ range with a resolution of 8 cm^−1^. A Vario MACRO cube elementary analyzer (Elementar Analysensysteme GmbH, Germany) was used to obtain an analysis of the CHNS elements and recalculate the number of functional groups (C_NN_) in the synthesized materials. A Zetasizer Nano ZS (Malvern, Great Britain) was utilized to measure the zeta potentials of sample suspensions (1 g/L concentration) in 1 mmol/L NaNO_3_ aqueous solutions in a range of pH values from 2 to 11 and to determine the isoelectric point (pI) at which the electrostatic repulsion forces ceased to act between the particles of the samples.

### 3.4. Adsorption Experiment

The batch procedure was applied to study the adsorption of Cu(II), Ni(II), and Eu(III) ions from their aqueous solutions and to determine the static sorption capacities (SSCs) of the synthesized samples (initial concentrations of C_0_(Cu^2+^), C_0_(Ni^2+^), or C_0_(Eu^3+^) = 0.25–4.5 mmol/L, sorbent dosage = 1 g/L, temperature = 25 °C, contact time = 24 h, ionic strength = 0.1 mol/L regulated by NaNO_3_). The pH of ~5.0–5.5 was created by the salts. The contents of Cu(II) and Ni(II) in the aqueous phase were determined using the atomic absorption method on a C-115-M1 spectrophotometer at the resonance lines of 324.7 nm (for Cu) and 232.0 nm (for Ni) in a depleted (oxide) flame of an acetylene/air mixture. The source of resonant radiation was an LS-2 spectral lamp. The concentrations of Eu(III) ions in the solutions were determined by titration with EDTA (with hexamethylenetetramine as a buffer and xylene orange as an indicator). The end of the titration was indicated by a change in the solution color from purple-pink to yellow.

## 4. Conclusions

The reaction of the co-condensation of TMPEDA with TEOS, BTESE, or BTESB in the basic environment created by ammonium hydroxide resulted in the formation of spherical organosilica particles (50 to 100 nm in size) with (propyl)ethylenediamine surface groups. It was established that the functional group content increased with a growing proportion of TMPEDA in the reaction mixture from 1.95 to 3.40 mmol/g (for TEOS-based polysiloxanes), from 1.5 to 3.4 mmol/g (for ethylene-bridged polysilsesquioxanes), and from 1.14 to 2.91 mmol/g (for phenylene-bridged polysilsesquioxanes). Meanwhile, the main condition for obtaining mesoporous samples was an increase in the proportion of TEOS or bissilanes in the reaction medium. 

It was found that both the polysiloxane and polysilsesquioxane samples were better for removing copper(II) than nickel(II) or europium(III) ions from aqueous solutions. The highest value of SSC relative to Cu^2+^ was characteristic of polysilsesquioxane with ethylenediamine groups synthesized at a BTESE/TMPEDA ratio of 1/1 (1.8 mmolCu/g, with C_NN_ = 3.4 mmol/g). As expected, the incorporation of ethylene bridges into the silica network prevented the N-containing groups from ‘back-sticking’, making them available for interactions with target cations.

Ni^2+^ ion uptake by the materials was apparently connected to their hydrophobicity, namely their ability to contribute to the destruction of nickel aquacomplexes. Thus, the phenylene-bridged polysilsesquioxane sample synthesized at a BTESB/TMPEDA ratio of 1/1 had the highest Ni^2+^ SSC value of 0.83 mmol/g among all the samples studied in the current article. 

The findings showed that during the adsorption of large ions such as Eu^3+^, forming eight-membered complexes, the number of functional groups in the material is important, but the location and the arrangement of these functional groups on the surface has a predominant role in the formation of the adsorption complex. Thus, europium(III) ions displayed the highest adsorption (SSC = 0.55 mmol/g) on the TEOS-based polysiloxane sample synthesized at a TEOS/TMPEDA ratio of 3/1. This mesoporous material, which had a high group content of 2.91 mmol/g, was formed by loosely packed spherical particles and had the highest S*_BET_* of 427 m^2^/g and a V*_p_* of 0.51 cm^3^/g.

The process of the adsorption of target cations by the samples that demonstrated the highest SSC values towards them had higher correlation coefficients with the Freundlich isotherm model, indicating that the highest adsorption was observed with the involvement of different adsorption centers on heterogeneous surfaces.

## Figures and Tables

**Figure 1 molecules-28-00430-f001:**
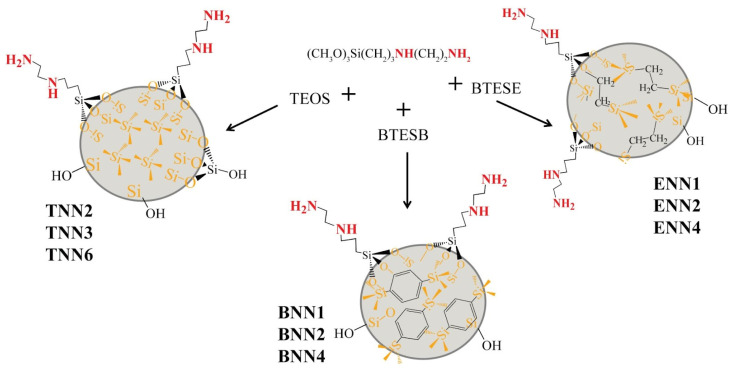
The synthesis scheme of organosilica adsorbents with (propyl)ethylenediamine groups on the surface.

**Figure 2 molecules-28-00430-f002:**
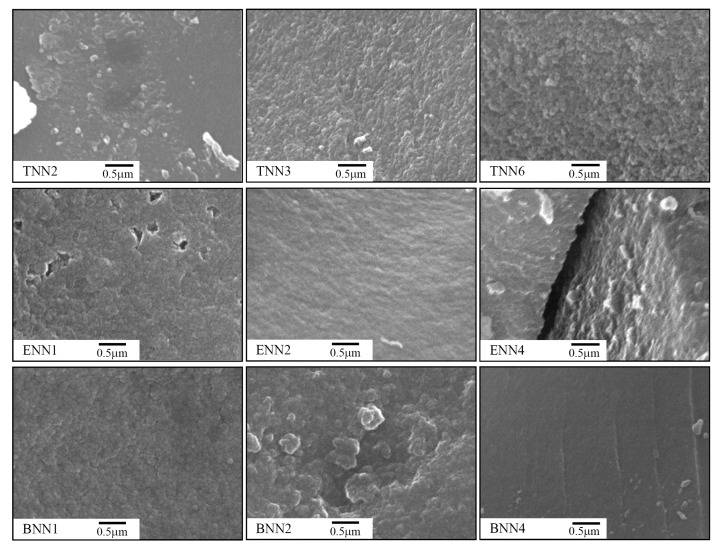
SEM images of the organosilica adsorbents with (propyl)ethylenediamine groups.

**Figure 3 molecules-28-00430-f003:**
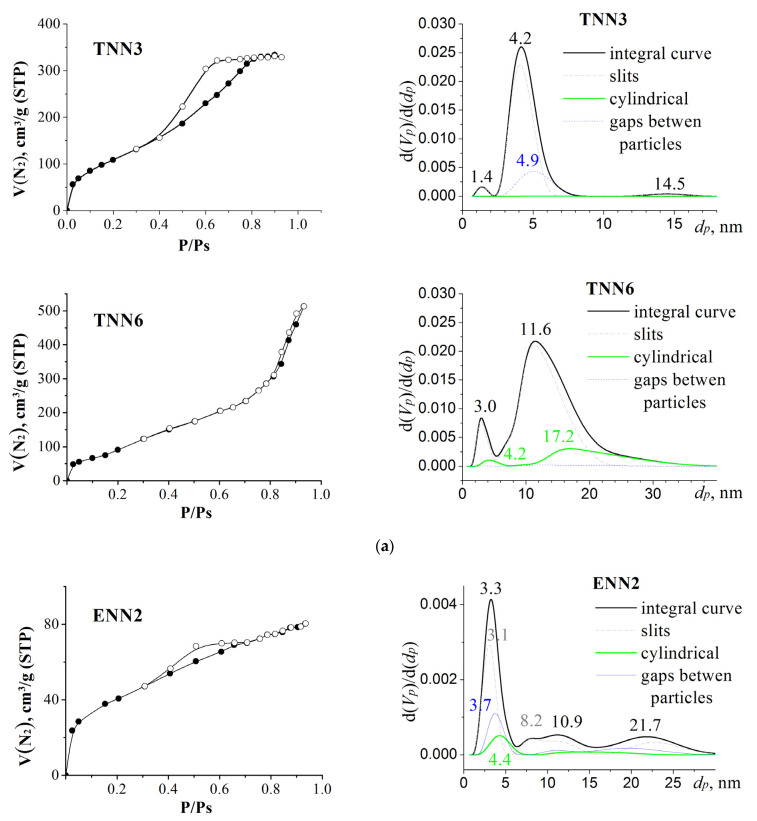
The low-temperature nitrogen adsorption–desorption isotherms collected for the (propyl)ethylenediamine-bearing samples and PSD plotted with the SCV/SCR method for: (**a**) TEOS-based polysiloxanes, (**b**) BTESE-based polysilsesquioxanes, and (**c**) BTESB-based polysilsesquioxanes.

**Figure 4 molecules-28-00430-f004:**
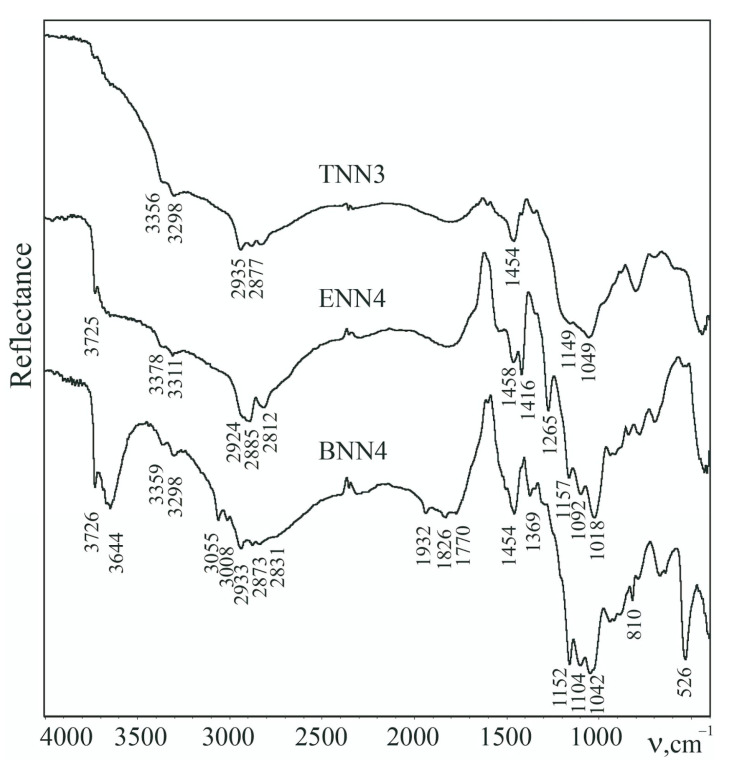
The DRIFT spectra of polysiloxane (**TNN3**) and polysilsesquioxane samples with ethylene (**ENN4**) and phenylene (**BNN4**) bridges, recorded at 200 °C.

**Figure 5 molecules-28-00430-f005:**
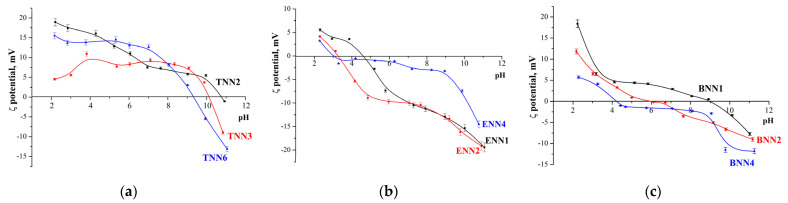
Zeta potential of polysiloxane TNN (**a**) and polysilsesquioxane ENN (**b**) and BNN (**c**) organosilica adsorbents with (propyl)ethylenediamine groups at different pH values.

**Figure 6 molecules-28-00430-f006:**
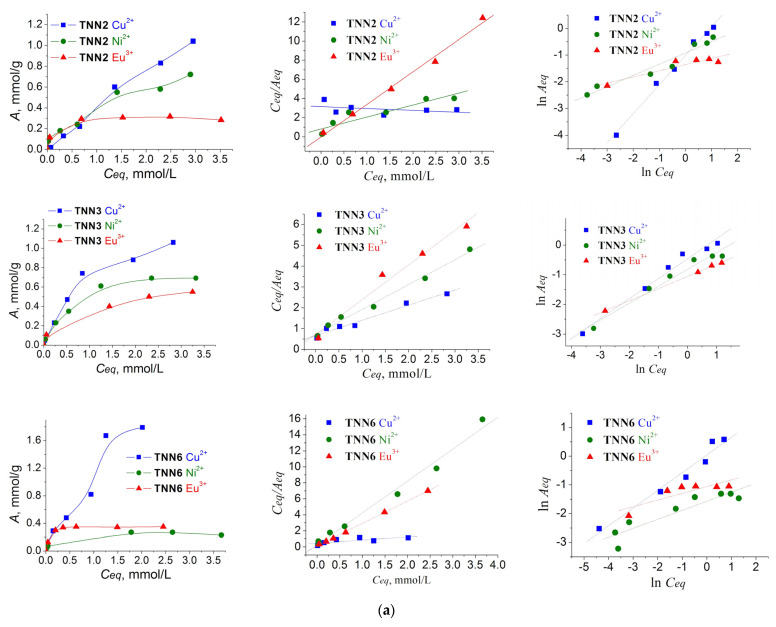
The isotherms of copper(II), nickel(II), and europium(III) ions adsorption by the ethylenediamine-bearing polysiloxane TNN (**a**) and polysilsesquioxane ENN (**b**) and BNN (**c**) organosilicas and their linearized graphs in the coordinates of the Langmuir (C*_eq_*/A*_eq_* vs. C*_eq_*) and Freundlich (ln A*_eq_* vs. ln C*_eq_*) isotherm equations.

**Figure 7 molecules-28-00430-f007:**
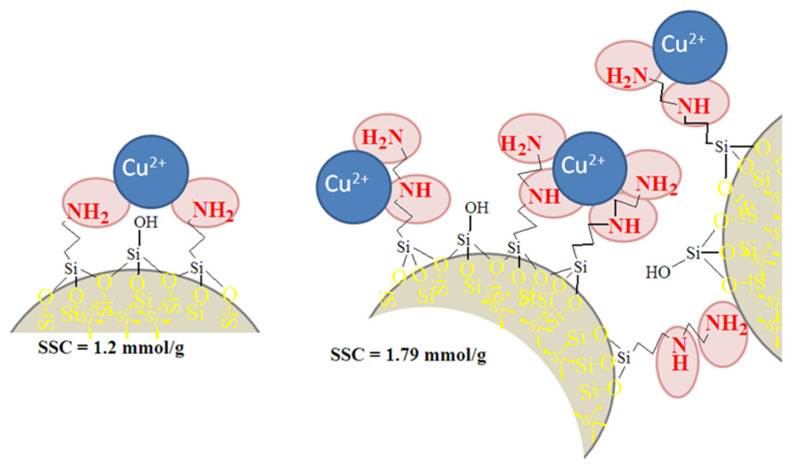
Expected scheme of copper(II) ion interaction with aminopropyl and ethylenediamine groups on the surface of TEOS-based polysiloxanes.

**Table 1 molecules-28-00430-t001:** Structural characteristics of samples, types of pores, and their contributions to the porous structures of materials.

Sample	Structuring Silane	Structuring to Functional Silane Molar Ratio	S*_BET_*, m^2^/g	V*_p_*, cm^3^/g	V*_pores_*, %			V*_pores_*, cm^3^/g		
					Nano	Meso	Macro	Slit-Shaped	Cylindrical	Gaps
**TNN2**	*Si(OC_2_H_5_)_4_*	2/1	2	-	-	-	-	-	-	-
**TNN3**	*Si(OC_2_H_5_)_4_*	3/1	427	0.51	7.0	93.0	0.0	0.423	0.000	0.085
**TNN6**	*Si(OC_2_H_5_)_4_*	6/1	400	0.79	2.8	95.9	1.3	0.674	0.112	0.008
**ENN1**	*≡Si–C_2_H_4_–Si≡*	1/1	12	-	-	-	-	-	-	-
**ENN2**	*≡Si–C_2_H_4_–Si≡*	2/1	147	0.12	10.3	89.7	0.0	0.081	0.014	0.028
**ENN4**	*≡Si–C_2_H_4_–Si≡*	4/1	343	0.27	14.6	85.4	0.0	0.115	0.065	0.089
**BNN1**	*≡Si–C_6_H_4_–Si≡*	1/1	190	0.21	1.1	98.9	0.0	0.042	0.092	0.072
**BNN2**	*≡Si–C_6_H_4_–Si≡*	2/1	7	-	-	-	-	-	-	-
**BNN4**	*≡Si–C_6_H_4_–Si≡*	4/1	400	0.26	28.6	70.6	0.8	0.117	0.115	0.027

**Table 2 molecules-28-00430-t002:** Characteristics of the synthesized ethylenediamine-bearing silica samples.

Sample	Structuring Silane	Structuring to Functional Silane Molar Ratio	N, % wt	C_NN_, mmol/g	pI	SSC Ni^2+^, mmol/g	SSC Cu^2+^, mmol/g	SSC Eu^3+^, mmol/g
**TNN2**	*Si(OC_2_H_5_)_4_*	2/1	8.62	3.40	10.77	0.72	1.04	0.32
**TNN3**	*Si(OC_2_H_5_)_4_*	3/1	7.54	2.91	10.14	0.69	1.06	0.55
**TNN6**	*Si(OC_2_H_5_)_4_*	6/1	5.17	1.95	9.32	0.27	1.79	0.35
**ENN1**	*≡Si–C_2_H_4_–Si≡*	1/1	8.79	3.40	4.59	0.46	1.80	0.40
**ENN2**	*≡Si–C_2_H_4_–Si≡*	2/1	6.19	2.36	3.27	0.34	1.62	0.37
**ENN4**	*≡Si–C_2_H_4_–Si≡*	4/1	4.02	1.50	3.03	0.24	0.70	0.25
**BNN1**	*≡Si–C_6_H_4_–Si≡*	1/1	7.54	2.91	9.19	0.83	1.14	0.52
**BNN2**	*≡Si–C_6_H_4_–Si≡*	2/1	5.01	1.88	5.41	0.22	1.38	0.17
**BNN4**	*≡Si–C_6_H_4_–Si≡*	4/1	3.10	1.14	4.11	0.34	0.58	0.32

**Table 3 molecules-28-00430-t003:** The parameters of Cu(II), Ni(II), and Eu(III) ion sorption by the organosilica adsorbents with (propyl)ethylenediamine groups in the coordinates of the Langmuir and Freundlich equations.

Sample	Cation	C_NN_, mmol/g	SSC, mmol/g	1Me/xN Formal Ratio	Langmuir Isotherm C*_eq_*/A*_eq_* = 1/(K*_L_* · A;*_max_*) + (1/A*_max_*) · C*_eq_*			Freundlich Isotherm lnA*_eq_* = lnK*_F_* + (1/n) · lnC*_eq_*		
					A*_max_*, mmol/g	K*_L_*, L/mmol	R^2^	n	K*_F_*, mmol/g	R^2^
**TNN2**	Cu(II)	3.40	1.04	1/17	0.39	0.155	**0.998**	16.18	0.062	0.809
	Ni(II)		0.72	1/3.0	1.78	5.022	**0.998**	1.78	1.530	0.900
	Eu(III)		0.32	1/13.7	0.40	1.521	**0.943**	1.76	0.203	0.854
**TNN3**	Cu(II)	2.91	1.06	1/13.0	0.31	0.097	**0.977**	14.79	0.068	0.301
	Ni(II)		0.69	1/4.3	0.80	2.464	**0.993**	1.79	0.499	0.863
	Eu(III)		0.55	1/12.5	0.24	3.353	**0.963**	5.29	0.168	0.168
**TNN6**	Cu(II)	1.95	1.79	1/12.0	0.27	0.071	**0.999**	−94.34	0.011	0.003
	Ni(II)		0.27	1/5.1	1.18	2.354	**0.926**	1.92	0.733	**0.951**
	Eu(III)		0.35	1/7.0	0.94	1.937	**0.957**	2.16	0.524	**0.972**
**ENN1**	Cu(II)	3.4	1.80	1/11.2	0.53	0.286	**0.995**	5.84	0.108	0.788
	Ni(II)		0.46	1/2.7	1.45	2.884	**0.970**	1.96	0.948	**0.973**
	Eu(III)		0.40	1/13.9	0.32	0.416	**0.399**	1.48	0.088	**0.793**
**ENN2**	Cu(II)	2.4	1.62	1/22.1	0.12	0.014	**0.994**	4.93	0.171	0.518
	Ni(II)		0.35	1/3.9	0.57	2.689	**0.908**	2.79	0.365	0.886
	Eu(III)		0.37	1/6.7	0.36	1.870	**0.924**	2.48	0.192	0.898
**ENN4**	Cu(II)	1.5	0.70	1/7.1	0.34	0.118	**0.997**	9.25	0.203	0.848
	Ni(II)		0.24	1/17	0.39	0.155	**0.998**	16.18	0.062	0.809
	Eu(III)		0.25	1/3.0	1.78	5.022	**0.998**	1.78	1.530	0.900
**BNN1**	Cu(II)	2.91	1.14	1/13.7	0.40	1.521	**0.943**	1.76	0.203	0.854
	Ni(II)		0.83	1/13.0	0.31	0.097	**0.977**	14.79	0.068	0.301
	Eu(III)		0.52	1/4.3	0.80	2.464	**0.993**	1.79	0.499	0.863
**BNN2**	Cu(II)	1.88	1.38	1/12.5	0.24	3.353	**0.963**	5.29	0.168	0.168
	Ni(II)		0.27	1/12.0	0.27	0.071	**0.999**	−94.34	0.011	0.003
	Eu(III)		0.17	1/5.1	1.18	2.354	**0.926**	1.92	0.733	**0.951**
**BNN4**	Cu(II)	1.14	0.58	1/7.0	0.94	1.937	**0.957**	2.16	0.524	**0.972**
	Ni(II)		0.34	1/11.2	0.53	0.286	**0.995**	5.84	0.108	0.788
	Eu(III)		0.32	1/2.7	1.45	2.884	**0.970**	1.96	0.948	**0.973**

A*_eq_* is the adsorption capacity at equilibrium, mmol/g; K*_L_* is the Langmuir constant, which characterizes the adsorption energy, L/mmol; C*_eq_* is the equilibrium concentration of metal ions in the solution, mmol/L; A*_max_* is the maximal adsorption capacity for a complete monolayer covering the surface, mmol/g; K*_F_* is the Freundlich constant, mmol/g; n is an empirical parameter related to the intensity of adsorption. Here bold marks higher value of correlation coefficient among the two models (Langmuir or Freundlich) for clearer visual perception.

## Data Availability

The data presented in this study are available on request from the corresponding author.

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
