# Peer review of "Diamine Groups on the Surface of Silica Particles as Complex-Forming Linkers for Metal Cations"

_molecules, 2023, doi:10.3390/molecules28010430_

Round 1

Reviewer 1 Report

Dear Authors, 

A few questions and notes:

1. In Figure 2, it would still make sense to use the same magnification for all imagines. 

2. Should be added the information on how the prepared samples with amine groups were stored, because amino groups on silica surface quite sensitive to the effects of moisture and CO2 (e.g. in the air at room temperature or in nitrogen or argon atmosphere).

3. The adsorption experiment was performed at pH 5.0-5,5. Are the amino groups basic and suitable for adsorption?

Sincerely,

Author Response

Dear Editor,

Dear Reviewers,

We appreciate the effort undertaken for the careful revision of the manuscript. All the comments and suggestions were taken into consideration, helping to specify important details and thus improve the quality of the manuscript. The corrections that were made are highlighted in the manuscript with yellow. The point-by-point responses to the comments are given below.

Reviewer 2 Report

The authors have prepared the spherical organosilica particles (50 to 100 nm in size) with (propyl)ethylenediamine surface groups. It was found that mesoporous samples better remove copper(II), than nickel(II) or europium(III) ions from aqueous solutions, which is interesting for the related readers. The experiments have been well done and the results are also presented, there, I could recommend its publication. however, I don't know why authors select these ions? how about the other ions such as Zn(II), Mn(II) and Co(II) in the water? Moreover, europium(III) is also significantly different from the 3d ions. Why authors discuss them together? the paper could be accepted after minor revision. 

Author Response

Dear Editor,

Dear Reviewers,

We appreciate the effort undertaken for the careful revision of the manuscript. All the comments and suggestions were taken into consideration, helping to specify important details and thus improve the quality of the manuscript. The corrections that were made are highlighted in the manuscript with yellow. The point-by-point responses to the comments are given below. Please see the attachment.
